# A Read-and-Select Framework for Zero-shot Entity Linking

**Zhenran Xu, Yulin Chen, Baotian Hu**[*] and **Min Zhang**
Harbin Institute of Technology (Shenzhen), Shenzhen, China
{xuzhenran, 200110528}@stu.hit.edu.cn
{hubaotian, zhangmin2021}@hit.edu.cn

## Abstract

Zero-shot entity linking (EL) aims at aligning entity mentions to unseen entities to challenge the generalization ability. Previous methods largely focus on the candidate retrieval stage and ignore the essential candidate ranking stage, which disambiguates among entities and makes the final linking prediction. In this paper, we propose a read-and-select (ReS) framework by modeling the main components of entity disambiguation, i.e., mention-entity matching and cross-entity comparison. First, for each candidate, the reading module leverages mention context to output mention-aware entity representations, enabling mention-entity matching. Then, in the selecting module, we frame the choice of candidates as a sequence labeling problem, and all candidate representations are fused together to enable cross-entity comparison. Our method achieves the state-of-the-art performance on the established zero-shot EL dataset ZESHEL with a 2.55% micro-average accuracy gain, with no need for laborious multi-phase pre-training used in most of the previous work, showing the effectiveness of both mention-entity and cross-entity interaction. Code is available at `https://github.com/HITsz-TMG/Read-and-Select`.

## 1 Introduction

Entity Linking (EL) is the task of aligning entity mentions in a document to their referent entity in a knowledge base (KB). Considering the example in Figure 1, given the sentence "Professor Henry Jones Sr is **Indiana Jones**'s father . . .", the mention **Indiana Jones** should be linked to the entity *Indiana Jones (minifigure)*. As a bridge that connects mentions in unstructured text and entities in structured KBs, EL plays a key role in a variety of tasks, including KB population (Ji and Nothman, 2016), semantic search (Blanco et al., 2015), summarization (Dong et al., 2022), etc.

Figure 1: An example of ZESHEL, the zero-shot entity linking dataset. The cross-encoder often encounters confusion when presented with candidates that share lexical similarities with the mention.

Previous EL systems have achieved high performance when a large set of mentions and target entities is available for training. However, such labeled data may be limited in some specialized domains, such as biomedical and legal cases. Therefore, we need to develop EL systems that can generalize to unseen entity sets across different domains, highlighting the importance of zero-shot EL (Sil et al., 2012; Vyas and Ballesteros, 2021).

Zero-shot EL task is proposed by Logeswaran et al. (2019) along with the ZESHEL dataset. There are two key properties in the zero-shot setting: (1) labeled mentions for the target domain are unavailable; (2) mentions and entities are only defined through textual descriptions, without additional resources (e.g., alias tables and frequency) which many previous EL systems rely on. Most zero-shot EL systems consist of two stages (Wu et al., 2020): (1) candidate retrieval, where a small set of candi-

---

[*] Corresponding author.

dates are efficiently retrieved from KB; (2) candidate ranking, where candidates are ranked to find the most probable one. So far, many methods have been proposed the first-stage retrieval (Sun et al., 2022; Sui et al., 2022; Wu et al., 2022). However, the candidate ranking stage, which makes the final prediction, is important yet less discussed. This paper focuses on the **candidate ranking stage**.

The cross-encoder, introduced by (Logeswaran et al., 2019) is a widely-used candidate ranking model that facilitates **mention-entity matching**. It takes as input the concatenation of the mention context and each entity description, and outputs whether or not the mention refers to the concatenated entity. However, since each entity is encoded *independently* with the mention, the cross-encoder lacks the capability to compare *all* candidates at once and select the most appropriate one. Figure 1 shows an error case of the cross-encoder. When ranking candidates that share a high degree of lexical similarity with the mention, the cross-encoder tends to assign high scores to both candidates. It may even incorrectly rank an entity higher than the correct one. Given the mention "Indiana Jones", the wrong *Indiana Jones (theme)* is rated higher than the gold *Indiana Jones (minifigure)*), highlighting the necessity for **cross-entity comparison**.

Motivated by the above example, we propose a read-and-select (ReS) framework, explicitly modeling both mention-entity matching and cross-entity comparison. The ReS framework consists of two modules: (1) In the reading module, we prepend prefix tokens (Li and Liang, 2021) to the concatenation of entity description and mention context, and obtain mention-aware entity representations, enabling mention-entity matching. (2) In the selecting module, the choice of candidates is framed as a sequence labeling problem, and all entity representations are fused together to enable cross-entity comparison. Unlike previous pipeline approaches that contain independently trained models (Wu et al., 2020), our reading module and selecting module share parameters, and the whole model is trained end-to-end.

We evaluate our approach on the widely-used ZESHEL dataset (Logeswaran et al., 2019). Based on the experimental results, our ReS framework achieves the state-of-the-art performance with a 2.55% micro-average accuracy improvement. Besides its performance advantage, ReS comes with another benefit: It does not need laborious multi-

phase pre-training used in most of previous work, showing its strong generalization ability. Further ablation study and case study demonstrate the effectiveness of cross-entity interaction.

Our contributions are summarized as follows:

- We propose a read-and-select (ReS) framework for candidate ranking stage of the zero-shot EL task, explicitly modeling the main components of disambiguation, i.e., mention-entity matching and cross-entity comparison.

- We propose a new framing of candidate ranking as a sequence labeling problem.

- We achieve the state-of-the-art result on the established ZESHEL dataset, showing the effectiveness of both mention-entity and cross-entity interaction.

## 2 Related Work

Entity linking (EL) bridges the gap between knowledge and downstream tasks (Wang et al., 2023; Dong et al., 2022; Li et al., 2022). There have been great achievements in building general EL systems with Wikipedia as the corresponding knowledge base (KB). Among them, the bi-encoder has been particularly successful: Two encoders are trained to learn representations in a shared space for mentions and entities (Gillick et al., 2019). However, the actual disambiguation is only expressed via a dot product of independently computed vectors, neglecting mention-entity interaction. To this end, Wu et al. (2020) attempt to add a candidate ranking stage via stacking a cross-encoder after the bi-encoder. Recent years have seen more Transformer-based ranking models obtaining promising performance gains (Barba et al., 2022; De Cao et al., 2021), but most of proposed works are built on the assumption that the entity set is shared among the train and test sets (Sun et al., 2022). In many practical cases, the train and test sets may come from different domain distributions and disjoint entity sets, emphasizing the necessity of zero-shot EL.

Zero-shot EL has recently attracted great interest from researchers. To enable progress on this task, Logeswaran et al. (2019) propose a dataset called ZESHEL, where mentions in the test set are linked to unseen entities without in-domain labeled data. Following this work, a number of methods that operate on ZESHEL have been proposed, but they only focus on the retrieval stage by

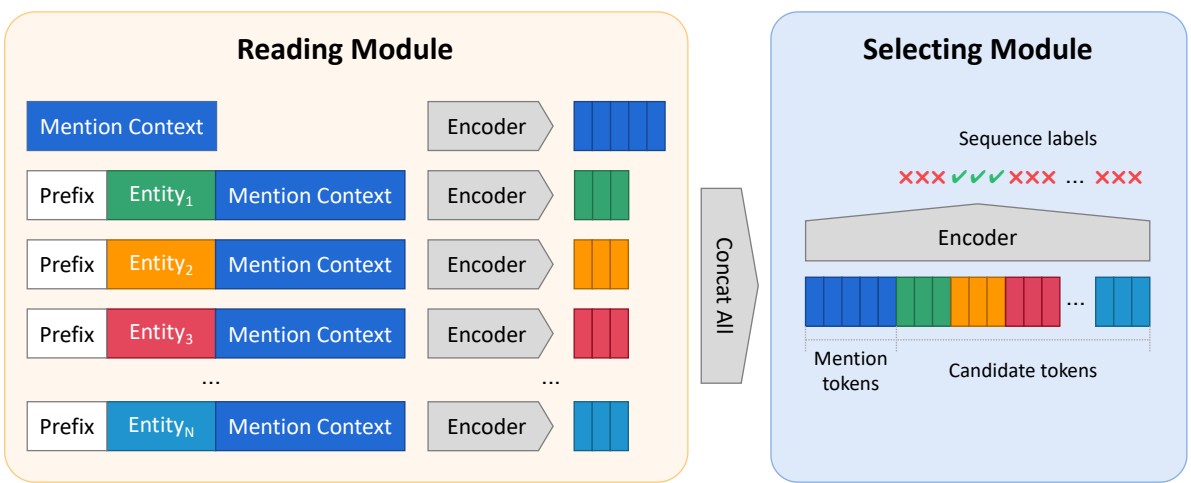

Figure 2: Illustration of our read-and-select (ReS) framework. Our reading module encodes mention context and uses prefix tokens as mention-aware entity representations. The above token embeddings are fused together and fed into the selecting module, where the entity is predicted through sequence labeling. The parameters are shared between the encoders in the reading and selecting module.

modifying the bi-encoder architecture, such as incorporating triplets in knowledge graphs (Ristoski et al., 2021), adding ultra-fine entity typing supervision (Sui et al., 2022), creating multi-view entity representations (Ma et al., 2021), and mining hard negatives for training (Zhang and Stratos, 2021; Sun et al., 2022). None of the above works considers the candidate ranking stage. However, the ranking stage is essential as it makes the final linking prediction among candidates, hence the need for further discussion about the ranking stage.

The few works focusing on the ranking stage all follow the cross-encoder formulation. Further improvements include extending to long documents (Tang et al., 2021; Yao et al., 2020) and different pre-training strategies (Logeswaran et al., 2019). In contrast, here we propose a sequence labeling formulation, where a model receives the mention, its context and all mention-aware entity representations as input, and marks out the the corresponding entity tokens. Note that this differs from the cross-encoder where each entity is independently encoded with mention context. With our framework, mention-entity matching and cross-entity comparison can be explicitly modeled.

## 3 Methodology

In this section, we describe the read-and-select (ReS) framework, our proposed method for zero-shot EL. As shown in Figure 2, the framework consists of the reading module and the selecting module, enabling mention-entity and cross-entity

interaction respectively. First, we formally present the task formulation in Section 3.1. Next, we introduce our reading module in Section 3.2. Finally, we introduce our selecting module and end-to-end training objective in Section 3.3.

### 3.1 Task Formulation

We first formulate the general entity linking (EL) as follows: given an entity mention $m$ in a document and a knowledge base (KB) $\mathcal{E}$ consisting of $N$ entities, i.e., $\mathcal{E} = \{e_1, e_2, ..., e_N\}$, the task is to identify the entity $e_i \in \mathcal{E}$ that $m$ refers to. The goal is to obtain an EL model on a train set of $M$ mention-entity pairs $D^{Train} = \{(m_i, e_i)|i = 1, ..., M\}$, which correctly labels mentions in test set $D^{Test}$. $D^{Train}$ and $D^{Test}$ are typically assumed to be from the same domain.

In this paper, we focus on the candidate ranking stage of zero-shot EL, formulated as follows: In the zero-shot setting, $D^{Train}$ and $D^{Test}$ contain $N_{src}$ and $N_{tgt}$ sub-datasets from different domains respectively, i.e., $D^{Train} = \{D_{src}^i|i = 1, ..., N_{src}\}$ and $D^{Test} = \{D_{tgt}^i|i = 1, ..., N_{tgt}\}$. Note that the entity sets in KBs (i.e., $\{\mathcal{E}_{src}^i|i = 1, ..., N_{src}\}$, $\{\mathcal{E}_{tgt}^i|i = 1, ..., N_{tgt}\}$) corresponding to the sub-datasets are disjoint. The entities and mentions are expressed as textual descriptions. In the candidate ranking stage, given the **mention** $m$ and its $k$ **candidate entities** (denoted as $Cnd(m) = \{e_1, ..., e_k\}$) from the first-stage candidate retrieval, the ranking model chooses the most probable entity in $Cnd(m)$ as the linking result.

## 3.2 Reading

The reading module aims to produce mention $m$'s representation and mention-aware representations of candidate entities $Cnd(m)$, in preparation for the input of the selecting module.

For the mention $m$, the input representation $\tau_m$ is the word-pieces of the mention and its surrounding context:

[CLS] $\text{ctxt}_l$ [START] $m$ [END] $\text{ctxt}_r$ [SEP]

where $\text{ctxt}_l$ and $\text{ctxt}_r$ are context before and after the mention $m$ respectively. [START] and [END] are special tokens to tag the mention. Then the *mention representation* (denoted as $\boldsymbol{H^m}$) is the token embeddings from the last layer of the Transformer-based encoder $T$:

$$\boldsymbol{H^m} = T(\tau_m) \in \mathbb{R}^{L_m \times d} \qquad (1)$$

where $L_m$ is the number of word-pieces in $\tau_m$ and $d$ is the hidden dimension.

For candidate entity $e$, the input representation $\tau_e$ is the word-pieces of its description. To obtain mention-aware entity representations, we prepend a PREFIX (Li and Liang, 2021) of length $L_p$ to the concatenation of entity description and mention context, and then feed into the encoder $T$. The output of the last layer can be computed as follows, enabling mention-entity matching:

$$\boldsymbol{H^e} = T([\text{PREFIX}; \tau_e; \tau_m]) \qquad (2)$$

We use the prefix's representation in $\boldsymbol{H^e}$ as the *mention-aware entity representation*, i.e.,

$$\boldsymbol{P^e} = [\boldsymbol{h}_1^e, ..., \boldsymbol{h}_{L_p}^e] \in \mathbb{R}^{L_p \times d} \qquad (3)$$

We write $\boldsymbol{h}_i^e \in \mathbb{R}^{1 \times d}$ to denote the $i$-th row of the matrix $\boldsymbol{H^e}$.

## 3.3 Selecting

We propose a selecting module with a new framing of candidate ranking as a sequence labeling problem. The model attends to mention context and all candidates together, and thus enables cross-entity comparison.

With the mention representations and mention-aware entity representations from the reading module, the selecting module concatenates them all, and feeds it into the encoder $T$. Note that the encoder here is the same as the one in the reading

module. The output of the last layer $\boldsymbol{H^{m,e}}$ can be denoted as follows:

$$\begin{aligned}\boldsymbol{H^{m,e}} &= T([\boldsymbol{H^m}; \boldsymbol{P^{e_1}}; ...; \boldsymbol{P^{e_k}}]) \\ &= [\boldsymbol{h}_1^m, ..., \boldsymbol{h}_{L_m}^m, \boldsymbol{p}_1^{e_1}, ..., \boldsymbol{p}_{L_p}^{e_1}, ..., \boldsymbol{p}_1^{e_k}, ..., \boldsymbol{p}_{L_p}^{e_k}]\end{aligned}$$
$$(4)$$

where $\boldsymbol{h}_i^m \in \mathbb{R}^{1 \times d}$, $\boldsymbol{p}_i^{e_j} \in \mathbb{R}^{1 \times d}$, and both notations represent a row in the matrix $\boldsymbol{H^{m,e}}$. Through self-attention mechanism (Vaswani et al., 2017) of the encoder, token-level interaction among candidates is conducted.

We frame the choice of candidates as a sequence labeling problem. The model learns to mark tokens of the corresponding entity *correct*, and mark other tokens *wrong*. The prediction score of the $i$-th token of candidate $e_j$, denoted as $\hat{s}_i^{e_j}$, is calculated through a classification head (i.e., a linear layer $\boldsymbol{W}$) with the sigmoid activation function:

$$\hat{s}_i^{e_j} = \text{Sigmoid}(\boldsymbol{W} \boldsymbol{p}_i^{e_j}) \qquad (5)$$

where $\hat{s}_i^{e_j} \in [0.0, 1.0]$, and a higher score means that the token $\boldsymbol{p}_i^{e_j}$ is more likely to be *correct*.

**Optimization.** For every mention $m$, we use the gold entity $e$ as the positive example, and use $N-1$ entities in $Cnd(m)$ ($e$ is not included) as negative examples. Suppose the candidate $e_j$ is the gold entity, all of its representation tokens (i.e., $\boldsymbol{p}_1^{e_j}, ..., \boldsymbol{p}_{L_p}^{e_j}$) should be labeled *correct*, and tokens of other entities should be labeled *wrong*. We optimize the encoder with a binary cross entropy loss:

$$\begin{aligned}\mathcal{L} = -\frac{1}{L_p \times N} \sum_{i=1}^{L_p} \sum_{j=1}^{N} (s_i^{e_j} \log(\hat{s}_i^{e_j}) \\ + (1 - s_i^{e_j})\log(1 - \hat{s}_i^{e_j}))\end{aligned} \qquad (6)$$

where $s_i^{e_j}$ takes the value 1 if the $\boldsymbol{p}_i^{e_j}$ token should be labeled *correct*, otherwise 0.

As the encoder $T$ in the reading and selecting module share parameters, our framework can be optimized end-to-end. Note that the concatenation order of candidate representations is random during training.

**Inference.** We obtain the final score for each candidate through a maximum pooling over all of its tokens:

$$\hat{s}^{e_j} = max(\{\hat{s}_1^{e_j}, \hat{s}_2^{e_j} ... \hat{s}_{L_p}^{e_j}\}) \qquad (7)$$

We use the highest score to choose the best candidate as the final linking result.

## 4 Experiment

In this section, we assess the effectiveness of ReS on candidate ranking of zero-shot EL. We first introduce the experimental setup we consider, i.e., the data (Section 4.1), evaluation metric (Section 4.2), technical details (Section 4.3), and baselines in comparison (Section 4.4). Then we present the main result of ReS, its category-specific results, and the ablation study in Section 4.5. More analysis about the cross-entity comparison can be found in the impact of candidate number (Section 4.6) and case study (Section 4.7).

### 4.1 Dataset

We conduct our experiments on the widely-used zero-shot EL dataset ZESHEL, which is constructed by Logeswaran et al. (2019) and built with documents from Wikia[1]. Table 1 shows the overall statistics of the dataset. In this dataset, multiple entity sets are available for training, with task performance measured on a disjoint set of test entity sets for which no labeled data is available. ZESHEL contains 16 specialized Wikia domains, partitioned into 8 domains for training, 4 for validation and 4 for test. The training set has a total of 49,275 labeled mentions while the validation and test sets both have 10,000 unseen mentions.

### 4.2 Evaluation Metric

Following previous work (Wu et al., 2020; Tang et al., 2021), we use the *normalized* accuracy as the evaluation metric. The *normalized* EL performance is defined as the performance evaluated on the **subset** of test instances for which the gold entity is among the top-$k$ candidates during the candidate retrieval stage. We use the dataset along with top-64 candidate sets retrieved by BM25[2]. The number of test instances is 1000, 974, 2785, 2053 for such **subsets** of the Forgotten Realms, Lego, Star Trek and YuGiOh domain respectively, resulting in a top-64 recall of 68% on the test sets. We compute average performance across a set of domains by both micro- and macro- averaging. Performance is defined as the accuracy of the single-best identified entity (top-1 accuracy).

| Domains | Entities | Mentions |
|---|---|---|
| **Training** | | |
| American Football | 31929 | 3898 |
| Doctor Who | 40281 | 8334 |
| Fallout | 16992 | 3286 |
| Final Fantasy | 14044 | 6041 |
| Military | 104520 | 13063 |
| Pro Wrestling | 10133 | 1392 |
| StarWars | 87056 | 11824 |
| World of Warcraft | 27677 | 1437 |
| **Validation** | | |
| Coronation Street | 17809 | 1464 |
| Muppets | 21344 | 2028 |
| Ice Hockey | 28684 | 2233 |
| Elder Scrolls | 21712 | 4275 |
| **Test** | | |
| Forgotten Realms | 15603 | 1200 |
| Lego | 10076 | 1199 |
| Star Trek | 34430 | 4227 |
| YuGiOh | 10031 | 3374 |

Table 1: Statistics of the zero-shot entity linking dataset ZESHEL.

### 4.3 Implementation Details

ReS is implemented with PyTorch 1.10.0 (Paszke et al., 2019). The encoder in the reading and selecting module share parameters, initialized with RoBERTa-base parameters (Liu et al., 2019). The number of parameters for ReS is roughly 124M.

ReS is trained on two NVIDIA 80G A100 GPUs. We use Adam optimizer (Kingma and Ba, 2015) with weight decay set to 0.01 for all experiments. Batch size is set at 4. The length of PREFIX $L_p$ is set at 3. We search learning rate among [5e-6,2e-5,4e-5] based on the validation set. The best-performing learning rate is 4e-5. We search the number of training candidates (i.e., $N$ in Equation 6) in [10,20,40,56] and the final number is set to 56 due to memory constraint.

For each mention context and each entity description in the reading module, we set the maximum length of the input at 256. We finetune for 4 epochs and choose the best checkpoint based on the micro-averaged normalized accuracy on the validation set. Finetuning ReS takes 8 hours per epoch.

---

[1] https://www.wikia.com
[2] The candidate sets are from the official ZESHEL Github repository: https://github.com/lajanugen/zeshel

| Method | Forgotten Realms | Lego | Star Trek | YuGiOh | Macro Acc. | Micro Acc. |
|---|---|---|---|---|---|---|
| Baseline (Logeswaran et al., 2019) | - | - | - | - | 77.05 | - |
| BLINK (Wu et al., 2020) | - | - | - | - | 76.58 | - |
| BLINK* (Wu et al., 2020) | 87.20 | 75.26 | 79.61 | 69.56 | 77.90 | 77.07 |
| GENRE* (De Cao et al., 2021) | 55.20 | 42.71 | 55.76 | 34.68 | 47.09 | 47.06 |
| ExtEnD* (Barba et al., 2022) | 79.62 | 65.20 | 73.21 | 60.01 | 69.51 | 68.57 |
| E-repeat (Yao et al., 2020) | - | - | - | - | 79.64 | - |
| Uni-MPR (Tang et al., 2021) | 87.25 | 78.57 | 80.56 | 67.31 | 78.42 | 76.65 |
| Bi-MPR (Tang et al., 2021) | **89.60** | **80.50** | 81.04 | 68.74 | 79.97 | 77.85 |
| ReS *(ours)* | 88.10 | 78.44 | **81.69** | **75.84** | **81.02** | **80.40** |
| ReS (w/o selecting) *(ours)* | 87.10 | 77.10 | 80.57 | 73.22 | 79.47 | 78.79 |

Table 2: Normalized accuracy of our ReS framework compared with previous state-of-the-art methods on test set of ZESHEL. **Bold** denotes the best results. "*" means our implementation. "-" means not reported in the cited paper.

## 4.4 Baselines

To evaluate the performance of ReS, we compare it with the following 7 state-of-the-art EL systems that represent a diverse array of approaches.

- Logeswaran et al. (2019) propose a baseline method based on cross-encoder. In Table 2, we report its best-performing variant, i.e., the cross-encoder with domain-adaptive pre-training (DAP), and denote this model as **Baseline**.

- **BLINK** (Wu et al., 2020) uses a cross-encoder based on BERT-base, which only finetunes on the training set of ZESHEL, without any pre-training phases. For a fair comparison with ReS, we implement a cross-encoder based on RoBERTa-base, denoted as **BLINK***.

- **GENRE** (De Cao et al., 2021) is an autoregressive method base on generating the title of the corresponding entity with constrained beam search.

- **ExtEnD** (Barba et al., 2022) proposes a new framing of entity disambiguation as a text extraction task;

- **E-repeat** (Yao et al., 2020) initializes larger position embeddings by repeating the small one from BERT-base, allowing reading more information in context. The best-performing variant also uses DAP.

- **Uni-MPR** (Tang et al., 2021) encodes a mention and each paragraph of an entity description, and aggregates these encodings through

an inter-paragraph attention mechanism. In addition, Uni-MPR also adopts DAP, and applies "Whole Entity Masking (WEM)" strategy in the masked language modeling (MLM) pre-training.

- **Bi-MPR** (Tang et al., 2021) is built on the above Uni-MPR. It adds another attention module to aggregate different paragraphs of the mention document. Same as Uni-MPR, Bi-MPR also uses WEM strategy in DAP.

## 4.5 Results

### 4.5.1 Main Results

We compare ReS with 7 previous state-of-the-art models in Section 4.4, and list the performance in Table 2. ReS outperforms all other models on ZESHEL, and improves previous best reported results by 1.05% macro-averaged accuracy and 2.55% micro-averaged accuracy, showing the effectiveness of our overall framework.

Besides its performance advantage, ReS comes with another benefit: it does not need laborious multi-phase pre-training. As stated in Section 4.4, Yao et al. (2020) and Tang et al. (2021) both use domain-adaptive pre-training (DAP) introduced by Logeswaran et al. (2019). Specifically, the model first pre-trains with texts in all domains (including source and target domains) with masked language modeling (MLM). Then for each target domain that the model is going to be applied on, an extra MLM pre-training stage is added before finetuning, using only the data in the target domain. This means that Yao et al. (2020) and Tang et al. (2021) use different checkpoints for different test domains. However,

ReS only needs one checkpoint for all domains and does not require any pre-training phase.

### 4.5.2 Domain-specific Results

We break down the overall performance into different test domains. As Table 2 shows, compared with BLINK* (better than the BLINK paper's reported result), our ReS framework outperforms it by 0.90, 3.18, 2.08, 6.28 accuracy points on the Forgotten Realms, Lego, Star Trek, YuGiOh domains respectively. Specifically, the improvement in the YuGiOh domain is the most significant. We hypothesize the reason is that the test domain "YuGiOh" is closely related to the train domains "Star Wars" and "Final Fantasy" since they all belong to the super-domain of comics. By reading related contexts and entity descriptions during training, ReS can effectively generalize to similar domains.

### 4.5.3 Category-specific Results

Logeswaran et al. (2019) categorize the mentions based on token overlap between mentions and the corresponding entity title as follows:

- *High Overlap (HO)*: mention string is identical to its gold entity title.

- *Multiple Categories (MC)*: The gold entity title is mention text followed by a disambiguation phrase (e.g., in Figure 1, mention string: "Indiana Jones", entity title: "Indiana Jones (minifigure)").

- *Ambiguous substring (AS)*: mention is a substring of its gold entity title (e.g., mention string: "Agent", entity title: "The Agent").

- All other mentions are categorized as *Low Overlap (LO)*.

According to the fact that the performance of candidate retrieval models in *Multiple Categories* and *Low Overlap* subsets is much lower than the other two subsets (Sui et al., 2022), we conjecture that mentions in *MC* and *LO* are more difficult and ambiguous, and require more complex reasoning. As Table 3 shows, ReS achieves the state-of-the-art performance in the *LO* subset. Compared with BLINK*, the improvement is the most notable in the *MC* subset with a 5.89% gain. This indicates that our ReS framework has the reasoning ability to compare candidates and choose the best one.

| Method | HO | MC | AS | LO |
|---|---|---|---|---|
| Baseline | 87.64 | 77.27 | 75.89 | 71.46 |
| BLINK* | 94.30 | 75.40 | **79.95** | 73.50 |
| Uni-MPR | 91.43 | 79.07 | 75.60 | 73.53 |
| Bi-MPR | 92.84 | **81.93** | 77.37 | 73.88 |
| ReS | **94.42** | 81.29 | 77.80 | **76.51** |
| ReS (w/o selecting) | 92.72 | 78.30 | 79.00 | 75.50 |

Table 3: Accuracy on the category-specific subsets including High Overlap (HO), Multiple Categories (MC), Ambiguous Substring (AS), Low Overlap (LO). "*" means our implementation.

### 4.5.4 Ablation Study

For the ablation of the **selecting module**, we consider a variant of ReS: remove the selecting module altogether, use the mention-aware entity representations (i.e., prefix tokens) from the reading module to obtain candidates' scores, thus without cross-entity interaction. Table 2 shows that removing the selecting module causes a performance drop across all test domains, leading to a 1.61% micro-averaged accuracy drop. Based on Table 3, compared with ReS (w/o selecting), ReS improves the most in the *Multiple Categories* subset. This indicates that cross-entity interaction is helpful in disambiguating lexically similar entities, which is in line with our motivation for fine-grained comparison among candidates illustrated in the case in Figure 1. More cases will be discussed in Section 4.7.

In addition, for the ablation of the **reading module**, we can compare the performance of BLINK* and ReS (w/o selecting): For mention-aware entity representations, BLINK* uses the [CLS] token while ReS uses the PREFIX tokens. According to Table 2, ReS (w/o selecting) outperforms BLINK* by an overall of 1.72% micro-averaged accuracy. Based on fine-grained results of 4 categories in Table 3, our prefix token embeddings achieve better performance on *Multiple Categories* and *Low Overlap* subsets, the aforementioned two relatively challenging subsets. These experimental results suggest that our reading module can better aggregate information from mention context and entity description, and thus create a better mention-aware entity representation.

### 4.6 Scaling the Number of Candidates

In Figure 3 (left), we report normalized accuracy with respect to the number of test candidates.

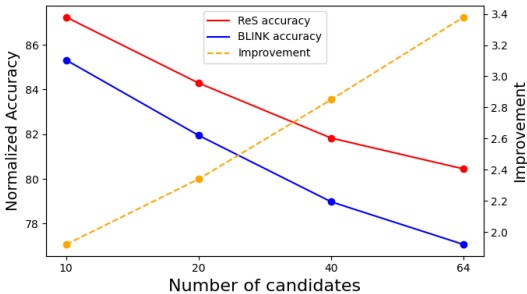 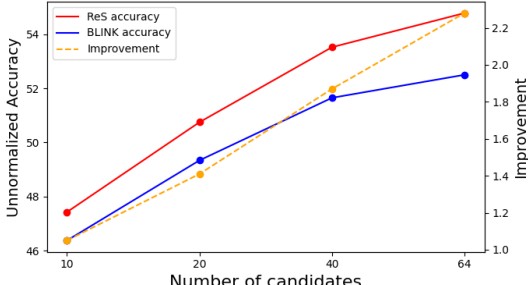

Figure 3: Micro-averaged normalized accuracy (left) and unnormalized accuracy (right) of our ReS framework and BLINK* on test sets as a function of the number of test candidates.

| Mention Context | ReS (w/o selecting) | ReS |
|---|---|---|
| It can hold onto objects with sticks like the **turkey leg**. Its LEGO Digital Designer name is TROLL LEFT HAND. | *Turkey*: A Turkey is a type of LEGO food. The pieces are in orange-brown and are made out of a rugged plastic material to show that it has been cooked. It is made up of Part 33048 and two of Part 33057. | ***Part 33057***: **Part 33057 is a piece shaped as a turkey leg. It can be held in a Minifigure's hand two ways. It is supposed to connect to Part 33048 to assemble a full turkey.** |
| Part 33013 is made to resemble a cake with fruit and chocolate on it. This part has only appeared in three sets, two Scala sets and one **Mickey Mouse** set. In the one Mickey Mouse set, it was used as a spare part. | *Mickey Mouse (figure)*: Mickey Mouse is a figure from the Mickey Mouse and the Disney's Baby Mickey theme. He appeared in all eight sets of the themes | ***Mickey Mouse (Theme)***: **Mickey Mouse is a LEGO theme based around Disney's Mickey Mouse and friends. The theme began and ended in 2000, and consisted of five sets.** |

Table 4: Examples of top 1 candidate predicted by ReS and its variant without selecting. **Bold** in mention contexts denotes mentions. The titles and descriptions of **gold entities** are in **bold**. The titles of entities are in *italics*.

Choosing among more candidates poses a challenge to candidate ranking models. As candidate number increases, the accuracy of both models decreases, but the accuracy of ReS decreases slower than BLINK*, leading to an increasing gap between their performance.

For evaluating end-to-end EL performance, we use *unnormalized* accuracy, which is computed on the entire test set. In Figure 3 (right), we can find that ReS consistently outperforms BLINK* for all candidate numbers. The improvement of ReS becomes larger as the candidate number increases.

The above two findings demonstrate that ReS is better than BLINK at aggregating information from more candidates, showing the effectiveness of cross-entity interaction.

### 4.7 Case Study

We list the candidate predicted by ReS and its variant without selecting in Table 4 for qualitative analysis. The examples are in *Low Overlap* and *Multiple Categories* subsets respectively. From these examples, We can infer that models with only mention-entity interaction (e.g., ReS without selecting) tend to confuse candidates which both have high lexical similarity with the mention context. However, with cross-entity comparison in the selecting module, ReS can make comprehensive judgements with contextual information and all candidates, and disambiguate better on such ambiguous cases.

## 5 Conclusion

In this work, we focus on the candidate ranking stage in zero-shot entity linking (EL). To disambiguate better among entities which have high lexical similarity with the mention, we propose a read-and-select (ReS) framework, which first produces mention-aware entity representations in the reading module, and then applies a sequence labeling paradigm on the fusion of all candidate representations in the selecting module, thus modeling both mention-entity and cross-entity interaction. Experiments show that ReS achieves the state-of-the-art result on ZESHEL, no need of any laborious pre-training strategy, showing its effective generalization and disambiguation ability. Future work may expand this framework to take longer contexts and descriptions into consideration.

## Limitations

Despite outperforming previous methods in macro and micro accuracy, our model does face limitations, chiefly due to the information loss when we restrict the length of mention context and entity descriptions to 256 tokens. The evidence for mention-entity matching could potentially reside in any paragraph within the entity document, rather than solely within the initial 256 tokens. This makes mention-aware entity representations less comprehensive, thereby impacting the interaction among candidates. Tackling the length constraint remains an intriguing avenue for future research.

## Acknowledgments

We thank Zifei Shan for discussions and the valuable feedback. This work is jointly supported by grants: Natural Science Foundation of China (No. 62006061, 82171475), Strategic Emerging Industry Development Special Funds of Shenzhen (No.JCYJ20200109113403826).

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
