# OpenReview forum: "A Read-and-Select Framework for Zero-shot Entity Linking"
_EMNLP/2023/Conference — EMNLP 2023 Findings_

### Official Review · Reviewer_WPJY · 2023-07-31

**Soundness:** 1

**Excitement:**

2: Mediocre: This paper makes marginal contributions (vs non-contemporaneous work), so I would rather not see it in the conference.

**Paper Topic And Main Contributions:**

This paper focuses the zero-shot entity linking task and proposes a framework to capture fine-grained cross-entity comparison information among candidates to improve the performance.

**Questions For The Authors:**

1. The T in Equation 1 and Equation 2 are the same Transformer-based encoder?

**Reasons To Accept:**

1. This paper tries to capture cross-entity comparison information for zero-shot entity linking task.
2. The method is easy to follow.

**Reasons To Reject:**

1. Poor figures (minor). Figures in this paper are not clear. I can not obtain the effectiveness of capturing fine-grained cross-entity interaction among candidates in comparison in Figure 1.
2. Poor motivation (major). The cross-encoder architecture is not "ignoring cross-entity comparison". It also "attends to all candidates at once" to obtain the final matching scores. Of course, it may be not so fine-grained.
3. Poor novelty of methodology (major). The idea of capturing fine-grained cross-entity interaction among candidates has already been proposed in entity linking (ExtEnD: Extractive entity disambiguation).
4. Unfair comparison (major). As far as I know, all previous zero-shot entity linking candidate ranking works use BERT-base parameters rather than RoBERTa-base and top 64 candidates rather than top 56. These may result in unfair comparison in the experimental results.
5. Wrong baseline results (minor). As far as I know, the results in BLINK and E-repeat are Micro Acc. rather than Macro Acc.

**Reproducibility:**

4: Could mostly reproduce the results, but there may be some variation because of sample variance or minor variations in their interpretation of the protocol or method.

**Reviewer Confidence:**

5: Positive that my evaluation is correct. I read the paper very carefully and I am very familiar with related work.

---

> ### Author Rebuttal · Authors · 2023-08-29
>
> Thanks for your time and constructive review! We address your concerns as follows:
>
> **Q1: Poor motivation**
>
> The cross-encoder focuses on **mention-entity interaction** to perform semantic matching between **a mention** and **an entity**, while our method attends to **all entities at one time** in the selecting module. We focus on **entity-entity interaction** to make comparisons among entities and capture their fine-grained difference.
>
> **Q2: Poor novelty of methodology**
>
> As stated in Line 110-112, our work is partly motivated by the promising cross-entity interaction. Though ExtEnD has explored such interaction and achieved state-of-the-art performance in general domain, **its performance drops under the zero-shot setting (shown in Table 2)**, significantly worse than the cross-encoder baseline and our method.
>
> In ExtEnD, the entity is only represented by **its name**, while our method use the reading module to obtain **mention-aware entity representations**, enabling more effective cross-entity comparison and more accurate prediction.
>
> **Q3: Unfair comparison**
>
> **Regarding the encoder:**
>
> Previous works based on BERT-base (Baseline, E-repeat, Uni-MPR and Bi-MPR) all utilize some continual pre-training strategies (e.g., domain-adaptive pre-training with in-domain corpus). However, our method does not need laborious multi-phase pre-training by simply applying RoBERTa-base. Our target is to learn how to compare candidates according to the their descriptions, and generalize across domains without the domain-specific corpus.
>
> In addition, as stated in Line 424-427, for fair comparison, we reproduced the cross-encoder based on RoBERTa-base, denoted as BLINK* in Table 2.
>
> **Regarding the number of candidates:**
>
> There might be a misunderstanding. As stated in Line 398-401, the number of **training** candidates is **56** due to memory constraint. The number of candidates during **inference** is **64**, same as the experimental setting in previous works.
>
> **Q4: Wrong baseline results**
>
> We double checked the BLINK paper and E-repeat paper, and confirmed that their reported results are computed by **macro-averaging**:
>
> - In the Table 2 of the BLINK paper: "Average performance across a set of worlds is computed by macro-averaging.".
> - In Section 4.1 of the E-repeat paper: "Average performance across a set of domains is computed by macro-averaging."
>
>
>
> **Q5: Poor figures**
>
> Figure 1 shows an error case of the cross-encoder, which motivates the fine-grained comparison among candidates. Thanks for your advice, and we will present the motivation more clearly in the next version.
>
> **Q6: The T in Equation 1 and Equation 2 are the same Transformer-based encoder?** Yes.
>
> We hope the above details and the general response have addressed your concerns. We are greatful for your help in improving the paper. If there are any additional questions or concerns, we are happy to engage in further discussions.

---

### Official Review · Reviewer_cEiX · 2023-08-03

**Soundness:** 3

**Excitement:**

3: Ambivalent: It has merits (e.g., it reports state-of-the-art results, the idea is nice), but there are key weaknesses (e.g., it describes incremental work), and it can significantly benefit from another round of revision. However, I won't object to accepting it if my co-reviewers champion it.

**Paper Topic And Main Contributions:**

The paper introduces a novel framework, termed Read-and-Select (ReS), designed to enhance the candidate ranking phase in zero-shot entity linking tasks. The ReS framework consists of two modules: a reading module and a selection module. The reading module generates mention-aware entity representations for each candidate entity, facilitating effective mention-entity matching. The selection module, on the other hand, amalgamates all candidate representations and produces sequence labels, thereby enabling a comprehensive cross-entity comparison. The proposed framework demonstrates superior reranking performance on the ZESHEL dataset.

**Reasons To Accept:**

- The paper enhances the candidate reranking phase of entity disambiguation, achieve new state-of-the-art performance on the ZESHEL dataset.
- The paper underscores the critical role of cross-entity comparison in the process, a factor that has been largely overlooked in prior research.

**Reasons To Reject:**

- The paper lacks comparative analysis with robust baselines, such as Bi-MPR. A detailed discussion highlighting the distinctions between this work and these baselines would have been beneficial.
- The concept of integrating entities in the encoder, although presented with variations, is not entirely novel and has been previously proposed in Fusion-in-decoder (https://arxiv.org/pdf/2007.01282.pdf).
- The scope of the task and dataset is somewhat limited, with a singular focus on the ZESHEL dataset. The paper could have been strengthened by reporting results on additional entity linking datasets, such as AIDA-YAGO.

**Reproducibility:**

5: Could easily reproduce the results.

**Reviewer Confidence:**

4: Quite sure. I tried to check the important points carefully. It's unlikely, though conceivable, that I missed something that should affect my ratings.

---

> ### Author Rebuttal · Authors · 2023-08-29
>
> Many thanks for the detailed and constructive review! We appreciate your positive feedback about our paper's motivation and novelty. Here, we address your main concerns:
>
> **Q1: The paper lacks comparative analysis with robust baselines, such as Bi-MPR.**
>
> Thanks for the insightful suggestion. We have introduced the difference of our method in Line 197-212, and compared the domain-specific and category-specific results with baselines (such as Bi-MPR) in Table 2 and Table 3. In the final version, we will ensure to highlight the discussion of the distinction between our method and baselines.
>
> **Q2: The concept of integrating entities in the encoder is not entirely novel and has been previously proposed in Fusion-in-decoder.**
>
> Our method differs significantly from Fusion-in-decoder (FID) in both **motivation** and **implementation**:
>
> - FID aims to **aggregate evidence** from multiple passages, while our method focuses on cross-entity interation to **make comparsions** among entities and capture their fine-grained difference.
> - FID simply concatenates **the whole sequence embeddings** of all passages, while our method employs **a prefix prompt** to obtain mention-aware entity representations before concatenation.
>
> In addition, we have tried to implement FID for zero-shot entity linking in our preliminary study, and encountered with the **out-of-memory (OOM)** problem with two 80G A100 GPUs, due to the long length of entity descriptions and mention contexts. Therefore, the FID's fusion strategy is not applicable for this task.
>
> **Q3: The scope of the task and dataset is somewhat limited, with a singular focus on the ZESHEL dataset.**
>
> As we focus on the problem of zero-shot entity linking, to the best of our knowledge, ZESHEL is the only well-established and widely-used benchmark for this task.
>
> Thanks for the suggestion of adding experiments on additional datasets. We have implemented our method on AIDA-YAGO (with the same experimental setting as ExtEnD [1]). The following table shows the results. Our method outperforms the state-of-the-art ExtEnD and exhibits strong performance in an *in-domain* setting.
>
> |    | AIDA Accuracy |
> |:---------:|--------------:|
> |   GENRE   |88.6|
> |   ExtEnD  |          90.0 |
> |    ours   |          **91.5** |
>
> *The results of GENRE and ExtEnD are taken from [1].
>
> We hope the above response and the general response have addressed your concerns, and we are grateful for your help in improving the paper. Please let us know if you have any further comments!
>
> Reference:
> [1] Edoardo Barba, Luigi Procopio, Roberto Navigli. 2022. ExtEnD: Extractive Entity Disambiguation. https://aclanthology.org/2022.acl-long.177/

---

### Official Review · Reviewer_kXSc · 2023-08-07

**Soundness:** 3

**Excitement:**

3: Ambivalent: It has merits (e.g., it reports state-of-the-art results, the idea is nice), but there are key weaknesses (e.g., it describes incremental work), and it can significantly benefit from another round of revision. However, I won't object to accepting it if my co-reviewers champion it.

**Paper Topic And Main Contributions:**

This paper presents a framework for zero-shot entity linking using mention-entity matching and candidate ranking as sequence labeling as key new contributions.

**Reasons To Accept:**

- (+) Simple, yet meaningful idea to improve candidate ranking for zero-shot EL.
- (+) Thorough comparison with previous work.
- (+) Competitive, state-of-the-art results.


**Reasons To Reject:**

- (-) Single, focused contribution: the paper is on the wordy side and could have been (imho...) easily presented as a short paper. I do not see enough experiments or analysis to justify a full-blown full paper.

**Reproducibility:**

4: Could mostly reproduce the results, but there may be some variation because of sample variance or minor variations in their interpretation of the protocol or method.

**Reviewer Confidence:**

4: Quite sure. I tried to check the important points carefully. It's unlikely, though conceivable, that I missed something that should affect my ratings.

---

### Meta-Review · Area_Chair_Zgs2 · 2023-09-18

**Recommendation:** 3

**Metareview:**

The paper introduces a method for reranking entity candidates for zero-shot Entity Linking.

All three reviewers acknowledge the importance of the cross-entity comparison for candidate ranking and Reviewers kXSc and cEiX also recognise the significant improvement to the state of the art by this work.

Reviewers kXSc and cEiX pointed out different ways in which the scope work and the manuscript itself are limited. The authors have responded positively to these points by providing new results and suggesting structural changes to the paper that would shift more original content (rather than background information) into the main paper. The issue regarding the scope pointed out by Reviewer cEiX (reranking in zero-shot with a single dataset) remains, but I think this mostly affects the excitement for rather than the soundness of the work.

After an extensive conversation with Reviewer WPJY, we were able to dill down to a few specific issues that they identified in the current work (poor motivation, low novelty, unfair comparisons). I agree with the reviewer on the latter - the change of encoder from BERT to RoBERTa means that previous works (other than the re-implemented BLINK) are at a disadvantage (and that an ablation study with BERT as an encoder would be a plus), but I don't think it's detrimental to the soundness of the work. Regarding the other two issues, my opinion is that they are not as severe as the reviewer presented them in their review and discussion and that neither impacts the soundness of the work.

---

### Decision · Program_Chairs · 2023-10-07

**Decision:**

Accept-Findings

**Comment:**

The paper introduces a method for reranking entity candidates for zero-shot Entity Linking.

All three reviewers acknowledge the importance of the cross-entity comparison for candidate ranking and Reviewers kXSc and cEiX also recognise the significant improvement to the state of the art by this work.

Reviewers kXSc and cEiX pointed out different ways in which the scope work and the manuscript itself are limited. The authors have responded positively to these points by providing new results and suggesting structural changes to the paper that would shift more original content (rather than background information) into the main paper. The issue regarding the scope pointed out by Reviewer cEiX (reranking in zero-shot with a single dataset) remains, but I think this mostly affects the excitement for rather than the soundness of the work.

After an extensive conversation with Reviewer WPJY, we were able to dill down to a few specific issues that they identified in the current work (poor motivation, low novelty, unfair comparisons). I agree with the reviewer on the latter - the change of encoder from BERT to RoBERTa means that previous works (other than the re-implemented BLINK) are at a disadvantage (and that an ablation study with BERT as an encoder would be a plus), but I don't think it's detrimental to the soundness of the work. Regarding the other two issues, my opinion is that they are not as severe as the reviewer presented them in their review and discussion and that neither impacts the soundness of the work.